# Determination of the Optimum Removal Efficiency of Fine Particulate Matter Using Activated Carbon Fiber (ACF)

**DOI:** 10.3390/ijerph17218230

**Published:** 2020-11-07

**Authors:** Min-Kyeong Kim, Yelim Jang, Duckshin Park

**Affiliations:** 1Future Innovation R&D Strategy Division, Korea Railroad Research Institute, Uiwang 16105, Korea; mkkim15@krri.re.kr; 2Transportation Environmental Research Team, Korea Railroad Research Institute, Uiwang 16105, Korea; yelimm412@krri.re.kr; 3Department of Transportation System Engineering, University of Science & Technology (UST), Daejeon 34113, Korea

**Keywords:** activated carbon fiber (ACF), adsorption efficiency, filter, fine particulate matter, railway tunnels

## Abstract

In Korea, concentrations of particulate matter (PM_10_) are significantly higher in urban railway tunnels (178.1 μg/m^3^) than in metropolitan areas (49 μg/m^3^). In railway tunnels in Korea, it was maintained at 3–4 times higher concentration than general atmosphere and platform. Dust generated by trains is scattered at high speed in these tunnels, making filtration difficult; therefore, the development of filters that can be maintained in tunnels is required. In the present study, we examined PM adsorption in the laboratory scale using activated carbon fiber (ACF), which has high adsorption and capacity. The ACF depth, velocity of flow, and fine PM concentration in the tunnel were the experimental variables. We compared PM concentrations before and after the filter experiments, and calculated removal efficiency to determine the optimal conditions. Comprehensive examination of the experimental variables and differential pressure showed that the optimal conditions for an ACF specimen were a wind speed of 3.0 m/s and the ACF depth of 400 mm. The average removal efficiency of PM_10_ was 55.5%, and that of PM_2.5_ was 36.6%. The reproducibility tests showed that the ACF filter could be washed and reused and is suitable for various places because it is easily maintained.

## 1. Introduction

Particulate matter (PM) is a key indicator of air pollution caused by natural and human activities; it affects air quality, regional and global climate, and human health [1,2,3,4]. Thus, the effects of fine PM on the human must be continuously studied. Several studies have reported effects of PM exposure on respiratory illness, heart failure, and mortality, especially PM with a diameter of <10 µm (PM_10_) or 2.5 µm (PM_2.5_). Significant increases in PM_2.5_ pollution have been reported; the risk of fatality following PM_2.5_ exposure is higher than that following PM_10_ exposure [5].

PM_10_ concentrations in urban railway tunnels (178.1 μg/m^3^) are significantly higher than those in metropolitan areas (49 μg/m^3^). In the case of the previous study [6], looking at the results of measuring all sections of the Seoul subway tunnel in 2015, the average concentration of PM_10_ was 98.0 ± 37.4, and when the frequency exceeding the PM_10_ maintenance standard was confirmed, lines exceeding 30% were found. In railway tunnels in Korea, it was maintained at 3–4 times higher concentration than general atmosphere and platform. The concentration of fine particulate in the tunnel can act as a factor of deteriorating urban air quality when discharged to the outside through ventilation [7]. The high PM concentration of the tunnel is caused by the inflow of polluted air from outside, rail wear, and the generation of fine particulate due to crushing of gravel and soil at the bottom of the tunnel. Dust generated by trains is scattered at train-induced airflow, making filtration difficult. Therefore, the development of a filter that can be easily maintained in tunnels is required.

Porous materials are widely used as adsorbents to remove harmful substances from the environment; among these, activated carbon is the most common, and is applied widely in water purification, food manufacturing, and solvent recovery, for example. However, since activated carbon has a long material adsorption time and a high dust generation rate, its use poses a risk of secondary contamination. In addition, the wide pore size distribution of activated carbon limits its capacity for adsorption separation of trace pollutants and selective adsorption of mixtures [8,9]. A previous study examined the heavy metal adsorption properties of granular activated carbon and activated carbon fibers (ACFs) and found that the ACFs exhibited high adsorption capacity [10].

To address these limitations, recent studies have examined the applicability of ACFs. ACFs are shaped like fiber yarn, with a diameter of ≤10 μm and specific surface area of 1200–3000 m^2^/g; ACFs with uniform micropores (10–20 Å) are currently in development. ACF has an adsorption capacity 1.5–10 times larger than that of activated carbon, and an adsorption rate that is 100–1000 times faster; moreover, the pressure loss is 1/4 to 1/10 that of granular activated carbon, allowing the removal and separation of trace substances. as ACFs are easily commercialized as nonwoven fabric due to its light weight and excellent flexibility and moldability [8], they are increasingly being applied within a range of fields. Micropores developed in fine fiber yarns provide rapid in-pore adsorption; unlike activated carbon, ACF performance is not dependent on internal diffusion resistance, such that adsorption devices can be miniaturized, are easily desorbed, and thus exhibit excellent regeneration properties. ACF density is lower than that of activated carbon, such that even a small amount can provide sufficient adsorption performance. Therefore, ACF is highly useful as an adsorbent for applications that require precise separation, such as adsorption water treatment in residential and industrial settings, air purification, and filtration of harmful exhaust gases [11,12,13].

In Japan, air pollution around arterial roads produced by automobile exhaust has emerged as a serious problem; local air purification and improved road structure and traffic conditions have been shown to be effective for reducing air pollution along roads. Roadside ventilation fences filled with ACF have been installed to remove NOx under natural wind power; NOx removal performance can be improved by washing the filters with water. This system represents an effective and environmentally friendly approach to reducing air pollution. The Japan Electric Power Research Institute is currently investigating similar NOx removal systems using ACF and fences, based on a panel and slit design [14,15,16]. The Korea Railroad Research Institute (KRRI) recently developed a functional ACF filter that can simultaneously remove fine PM and harmful microorganisms through the lamination of ACF with copper particles [17]. Some ACF studies have sought to remove volatile organic compounds (VOCs) such as benzene from polluted air along roads; the installation of an ionizer containing a carbon fiber electrode at the front end of the filter is one approach used to study the antibacterial and dust collection efficiency of ACF filters [18,19,20,21,22,23,24,25,26].

However, few studies have examined the efficiency of PM_2.5_ reduction using ACF filters. Therefore, in the present study, to further develop ACF as a filter for air purification, we applied an ACF developed for use in a roadside air filtration fence to determine the conditions under which the ACF removed fine PM, at concentrations similar to those observed in tunnels, with maximum efficiency.

## 2. Materials and Methods

### 2.1. Experimental Design

This lab experiment was performed from July to August 2020, and the average temperature at this time was 23.6 ± 1.9. To evaluate the removal efficiency of fine PM, including PM_2.5_, it used A1 ultrafine test dust (ISO 12103-1; particle diameter, 0–10 µm) as the target. A1 ultrafine test dust has a maximum silicon content of 69–77%, and is composed of aluminum, iron, sodium, calcium, magnesium, titanium, and potassium. In this experiment, we used ACF (Beihai Fiberglass Co., Ltd., Jiujiang, China) as a flexible, nonpowered filter, with a thickness of 3 mm and specific surface area of 1696 m^2^/g. The total pore volume of ACF is measured 0.09 cm^3^/g and the average pore size is measured 2.83 nm at BJH desorption. At this time, the pretreatment was performed at 110 °C for 4 h under vacuum conditions and TristarII 3020 (Micromeritics, Norcross, GA 30093-2901, USA) was used as the analyzer.

The ACF filter was attached to both sides of each panel of the test specimen, and the panel was erected vertically such that ultrafine dust could adhere to both sides (Figure 1).

The external and internal of the test specimen is white acrylic, and the material of the panel inserted into the test specimen is polycarbonate. The external size is 687 (W) × 698 (D) × 600 (L) mm, the internal size is 491 (W) × 491 (D) × 600 (L) mm, and the panel size is 500 (W) ×2 (D) × 600 (L) mm (Table 1).

After spraying A1 ultrafine test dust into the duct system using a particle disperser, the concentration of fine PM before and after filtration was measured by passing the ACF through the duct system (700 (W) × 700 (D) × 7135 (L) mm). While A1 ultrafine test dust was sprayed at a constant concentration, we measured the PM concentration at the before passing filter and at after passing the filter for 15 min at the same time. We repeated this process three times in each variable to determine the reproducibility of the results. To determine the PM removal efficiency of the ACF filter, ACF depth, wind speed, and fine PM concentration were considered as experimental variables.

To examine the effect of ACF depth on the PM filtration efficiency of the ACF, it varied the depth (200, 400, or 600 mm). To examine the effect of wind speed on fine PM filtration efficiency at each ACF depth, it varied the speed (0.1, 0.5, 1.0, 2.0, 3.0, or 5.0 m/s) (Table 1). Wind speed is a major variable in removing fine PM, and subway tunnels are semi-enclosed spaces, and train winds with high velocity are generated by the operation of trains. In the previous study [27], the train wind was generally on average 0–10 m/s when staring and entering the train. This study aims to apply not only to the subway tunnel but also to the roadside. Therefore, the average wind speed standard for each city in Korea was also reflected, and the variable was set at 0.1–5m/s.

At lower filter differential pressure loss rates, energy efficiency is higher when a power source is applied; therefore, since ACF is a nonpowered filter, it examined the differential pressure in the ACF for each wind velocity.

### 2.2. Experimental Equipment and Measurement

The fine PM measurement instrument used in this study were two pressure-independent aerosol spectrometers (11-s; Grimm Aerosol Technik, Berlin, Germany) that measured the concentration and number of fine particles (diameter, 0.25–32 µm) every 6 s, with 31 channels for each size. Additionally, the correlation analysis between the two pressure-independent aerosol spectrometers was reviewed.

This instrument is widely used for real-time dust concentration measurement [7]. We took the average filtration efficiency (measured every 6 s during a 15-min period) as the final filtration efficiency of the test specimen. To evaluate the fine PM removal efficiency of the ACF, it measured the filter differential pressure using a differential pressure gauge (Testo 400; Testo Korea Ltd., Bangkok, Thailand), and the filtration flow rate using a flow velocity meter (Air Velocity Transducer 8455; TSI, Shoreview, Minnesota 55126, USA).

The reactor (Anytech Co., Suwon, Korea) and test specimen used to evaluate the A1 ultrafine test dust removal efficiency of the ACF are shown in Figure 2. Using a dust generator, A1 ultrafine test dust was sprayed into the reactor. The reactor measures the fine PM concentration according to location, and analyzes fine PM propagation according to wind speed; it can reproduce wind speeds of 0–30 m/s, and allows the user to vary duct length and sample the PM concentration in the duct.

The test specimen was made of white acrylic according to the horizontal and vertical dimensions of the reactor (Anytech Co., Suwon, Korea). Since the ACF is flexible, the test specimen was manufactured in the form of a panel support. Taking into account the thickness and spacing of the ACF and government specifications, 19 panels were fabricated; the ACF was attached to both sides of each panel. The spacing between filters was 10 mm. The ACF depth was treated as an experimental variable; in experiments where part of the ACF was removed, electrostatic spray was applied to areas of the panels that would otherwise contain ACF.

## 3. Results and Discussion

### 3.1. PM Removal Efficiency of the ACF Filter

At an ACF depth of 200 mm, the average filtration efficiency was 5.9% for PM_10_ and 10.6% for PM_2.5_ (Table 2). At a flow rate of 1.0 m/s, the filtration efficiency was 10.5% for PM_10_ and 12.8% for PM_2.5_, indicating relatively high efficiency. At wind speeds of 1.0–3.0 m/s, filtration efficiency was above average for both PM_10_ and PM_2.5_. At extremely high and low flow rates, PM removal efficiency was relatively low. At this time, the removal efficiency was highest at 1 m/s, and the result was decreased as the wind speeds gradually increased. It is judged that the area where ACF is attached to the test specimen is as small as 200 mm, and the adsorption performance decreases as the wind speeds increases gradually (Table 2).

At an ACF depth of 400 mm, the average filtration efficiency was 41.1% for PM_10_ and 29.6% for PM_2.5_ (Table 3). At a flow rate of 3.0 m/s, the filtration efficiency was 50.5% for PM_10_ and 36.6% for PM_2.5_, indicating relatively high efficiency. At wind speeds of 2.0–5.0 m/s, the filtration efficiency was above average for both PM_10_ and PM_2.5_. Thus, when the ACF depth was doubled from 200 to 400 mm, the average PM_10_ value increased 7-fold, and the average PM_2.5_ value increased about 3-fold. Therefore, the filtration efficiency increased as the specific surface area of the ACF increased. At extremely low flow rates, the removal efficiency was relatively low, and when the flow rates exceeded 2.0 m/s, the removal efficiency was higher than average.

At an ACF depth of 600 mm, the average filtration efficiency was 44.3% for PM_10_ and 36.3% for PM_2.5_ (Table 4). At a flow rate of 5.0 m/s, the filtration efficiency was 65.9% for PM_10_ and 51.9% for PM_2.5_, indicating relatively high efficiency. Similar to our observation at 400 mm, at wind speeds of 2.0–5.0 m/s, the filtration efficiency was higher than average, especially above a certain flow velocity. The average fine PM filtration efficiency was slightly higher at an ACF depth of 600 mm than at 400 mm. The maximum average efficiency for each wind speed was greater than 50% for both PM_10_ and PM_2.5_ at an ACF depth of 600 mm.

The efficiency of the on-powered ACF filter did not reach a high level at flow rates of 0.1 and 0.5 m/s, regardless of ACF depth. The average PM removal efficiency was high at flow rates of 2.0–3.0 m/s.

Wind speed is a major variable that has an important influence on the amount of PM generated and removed [23]. In the case of a weak wind speed, the flow of PM was not affected, and the PM did not reach up to the ACF, making it difficult to adsorb. In addition, in the case of high wind speed, it is difficult to be adsorbed by ACF due to an unstable condition that quickly passes through the duct. In addition to the wind speed, the cross-section area of the filter can be seen as a very important factor, such as the ACF depth condition examined in this study.

Through varying ACF depth and wind speed, it found that an ACF depth of 400 mm provided the lowest differential pressure, especially at wind speeds of 2.0–3.0 m/s. The differential pressure was generally affected by wind speed. When the wind speed increases, the air volume increases and the area decreases. In this case, the friction loss increases, the differential pressure increases, and the collection efficiency of the filter decreases. In this experiment, in a specimen with a total depth of 600 mm, when an ACF of 400 mm was attached, the differential pressure was low at 2–3m/s, and it is judged that there is a correlation with the result that the removal efficiency was high (Table 5). In a previous study, the pressure loss due to a 2.5-mm-thick ACF with a specific surface area of 1350 m^2^/g and flow rate of 1.0 m/s was 200 Pa [16]. Thus, the test specimen examined in this study was relatively effective in lowering the differential pressure.

### 3.2. Optimal Efficiency and Local Government Applications

Our comprehensive examination of ACF depth, wind speed, differential pressure, and fine PM removal efficiency demonstrated that the optimal conditions for real-life application of the ACF test specimen used in this study were an ACF depth of 400 mm and flow rate of 3.0 m/s (Figure 3).

In a previous study that examined a slit-type paneled ventilated fence design, the NO and NO_2_ removal efficiency rates were significantly different versus without wind [15]. However, in the present study, filtration efficiency was examined using a vertical panel to remove any effect of differences in efficiency between the top and bottom of the panel. In that study [15], NO and NO_2_ removal efficiency rates were compared at wind speeds of 0.5 and 1.0 m/s, and NO removal efficiency was found to increase as wind speed decreased; at low wind speeds, NOx removal was promoted by adsorption and oxidation reactions. By contrast, in the present study, it targeted fine PM and observed high fine PM removal efficiency at moderate wind speeds of about 3.0 m/s. It may be that large amounts of fine PM are unable to reach the filter at low wind speeds, and that PM is not easily adsorbed at high wind speeds. The wind speed is a major factor in the PM removal efficiency, but the surface area is also a factor. In this experiment the ACF depth is not less than 200 mm, it can be confirmed the average efficiency of about 30% or more even at a low or high wind speed is seen based on PM_10_.

Based on the ultrafine dust removal efficiency results for the filter specimen examined in this lab experiment, it investigated the possibility of ACF filter application at an actual site. If natural wind occurs at the roadside, the ACF can function as an effective nonpowered filter that may be washed with water, making it an effective solution in terms of cost and environmental impact. We selected Simgokcheon, Korea (an ecological river restoration site) as the study area, wherein continuous natural wind is maintained. Wind speed was measured at a height of about 1m from the road in order to check the airflow generated by the contact between the car wheel and the road on the road. A road runs alongside the river, creating an ideal location for installation of the nonpowered filter. To confirm the suitability of the study site, we measured wind speed in the presence and absence of road traffic (Figure 4), and found that the average wind speed during traffic was 0.6 m/s (maximum = 3.47 m/s). Therefore, the test specimen was suitable for application at the study site.

## 4. Conclusions and Discussion

In this study, the efficiency of an adsorption method for removing fine PM was examined, using ACF as the adsorption material. We determined the optimal conditions for maximum fine PM removal efficiency by the ACF filter.

ACF has a large specific surface area, good adsorption rate, and high adsorption capacity; due to its light weight and excellent flexibility and moldability, it is easily commercialized as a nonwoven fabric. Accordingly, ACF has many applications, including water treatment, NOx and VOC removal from air. In this study, the efficiency of removing fine particulate of ACF was reviewed, and as ACF is a nonpowered filter, it is determined that it can contribute to the establishment of clean zone of fine particulate in roads and underground spaces by utilizing the wind generated by the movement of a car or train. We designed a prototype ACF installed in a vertical panel for use at an actual site to facilitate roadside air cleaning. Our results showed that the optimal conditions for this system were an ACF depth of 400 mm and air flow speed of 3.0 m/s. It examined the efficiency of the nonpowered filter in the laboratory and at an actual site (Simgokcheon, Korea), which has a continuous wind speed of about 3.0 m/s, similar to the optimal air flow speed of the proposed system. Therefore, the ACF filter was determined to be suitable not only for underground spaces, but also for roadside air purification as a measure to reduce the concentration of PM emitted by vehicles on roads. In addition, when the ACF filter is installed, in the case of the roadside, it will be possible to additionally perform a function in terms of environmental aesthetics by printing on the ACF. Additionally, it is significant in that it is not affected by the season or rain in terms of removing fine particulate generated by train winds according to train operation by installing it in an underground tunnel space. In addition, in the case of ACF filter, it can be applied without an additional power source due to the influence of airflow generated from roads and underground lines, thus has an advantage in economic terms.

In this study, there is a limitation in that only one site in Simgokcheon (Korea) was selected as the field application site of the ACF filter and the application was reviewed. In the future, as a target site considering various environmental factors that can apply the advantages of the ACF filter, we plan to additionally review underground spaces and roadside spaces, and conduct comparative analysis.

In a future study, we will investigate the optimal conditions under which ACF filters can reduce the effects of NOx on the generation of particulate pollutants, using a combination of laboratory and field experiments.

## Figures and Tables

**Figure 1 ijerph-17-08230-f001:**
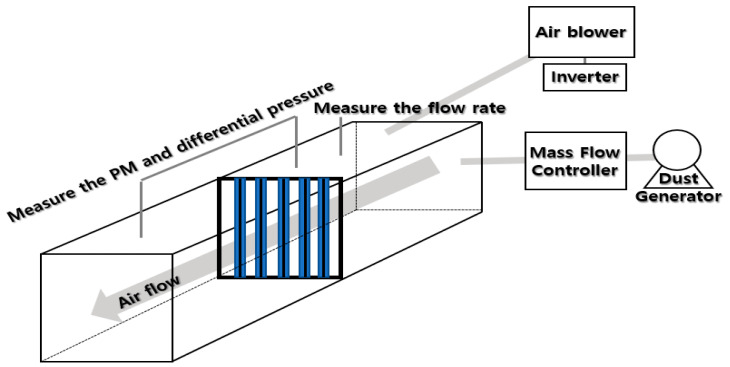
Activated carbon fiber (ACF) system used to evaluate fine particulate matter (PM) filtration performance in this study.

**Figure 2 ijerph-17-08230-f002:**
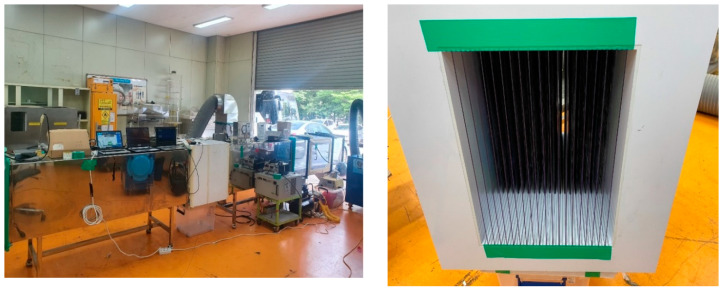
The reactor and test specimen used for the ACF efficiency measurements.

**Figure 3 ijerph-17-08230-f003:**
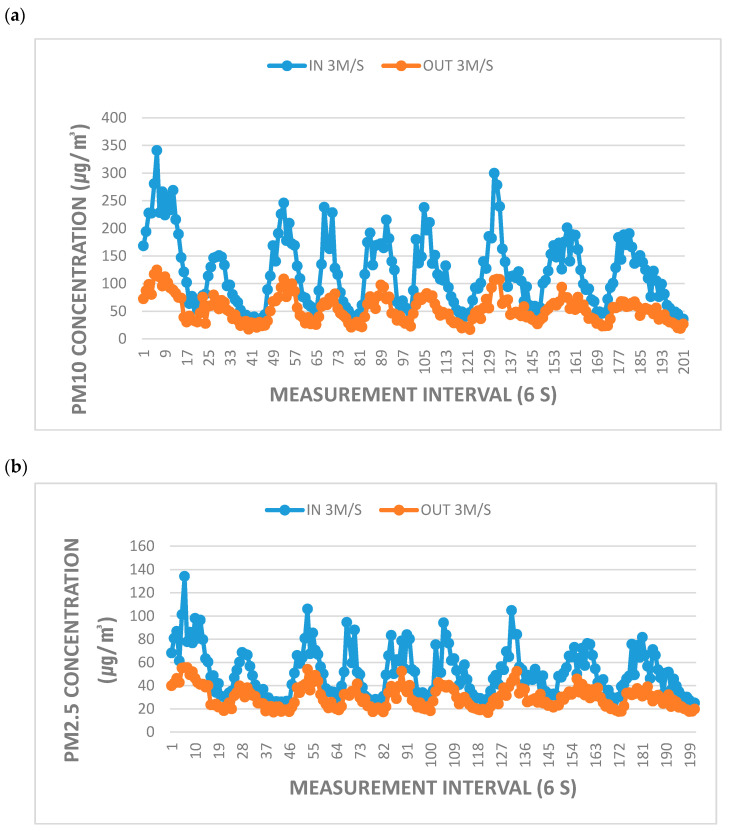
Variation in fine PM removal efficiency under optimal conditions: an ACF depth of 400 mm and flow rate of 3.0 m/s; (**a**) PM_10_; (**b**) PM_2.5_.

**Figure 4 ijerph-17-08230-f004:**
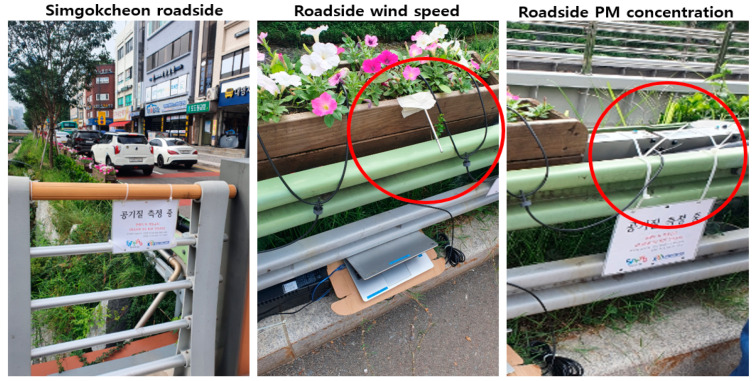
Setup for measuring wind speed and fine PM concentration in the ACF filter in the study area.

**Table 1 ijerph-17-08230-t001:** Summary of the activated carbon fiber (ACF) performance experiment.

Particulate Matter		A1 Ultrafine Test Dust (ISO 12103-1)
Reactor	Duct	Air flow generator (Fine PM propagation path analysis product)
	Dimensions (in mm)	700 × 700 × 7135
Test specimen	Material	(External/Internal) White acrylic (Panel) Polycarbonate
	Dimensions (in mm)	(External) 698 × 698 × 600 (Internal) 491 × 491 × 600 (Panel) 500 × 2 × 600
ACF	Thickness (mm)	3
	Depth (mm)	200, 400, and 600
	Interval (mm)	10
Wind speed (m/s)		0.1, 0.5, 1.0, 2.0, 3.0, and 5.0

**Table 2 ijerph-17-08230-t002:** Average fine particulate matter (PM) removal efficiency of the ACF filter for different wind speeds at an ACF depth of 200 mm.

Wind Speed (m/s)	PM_10_			PM_2.5_		
	Before Filtration (μg/m^3^)	After Filtration (μg/m^3^)	Removal Efficiency (%)	Before Filtration (μg/m^3^)	After Filtration (μg/m^3^)	Removal Efficiency (%)
0.1	97.5 ± 6.1	92.9 ± 4.8	4.9	58.7 ± 6.0	53.5 ± 6.9	8.8
0.5	101.2 ± 9.7	100.5 ±17.1	0.7	64.4 ± 9.7	58.3 ± 11.2	9.4
1.0	102.8 ± 4.7	91.9 ± 8.7	10.5	60.2 ± 7.6	52.5 ± 7.5	12.8
2.0	105.5 ± 7.2	96.2 ± 11.2	8.8	61.1 ± 7.9	53.6 ± 7.3	12.3
3.0	104.9 ± 3.8	97.2 ± 4.9	7.4	58.3 ± 4.4	51.7 ± 3.3	11.3
5.0	104.5 ± 8.2	101.1 ± 7.7	3.2	53.2 ± 5.4	48.3 ± 4.1	9.2

**Table 3 ijerph-17-08230-t003:** Average fine PM removal efficiency of the ACF filter for different wind speeds at an ACF depth of 400 mm.

Wind Speed (m/s)	PM_10_			PM_2.5_		
	Before Filtration (μg/m^3^)	After Filtration (μg/m^3^)	Removal Efficiency (%)	Before Filtration (μg/m^3^)	After Filtration (μg/m^3^)	Removal Efficiency (%)
0.1	88.1 ± 27.7	57.5 ± 11.2	34.7	50.3 ± 17.2	36.8 ± 10.3	26.9
0.5	108.8 ± 13.3	77.7 ± 10.7	28.6	48.2 ± 10.5	38.9 ± 7.4	19.2
1.0	113.2 ± 10.9	70.2 ± 7.9	38.0	51.0 ± 10.0	37.8 ± 6.5	25.9
2.0	112.4 ± 5.7	59.1 ± 3.5	47.4	52.1 ± 9.6	34.9 ± 6.6	33.1
3.0	118.4 ± 7.2	58.5 ± 5.0	50.5	56.5 ± 5.5	35.8 ± 5.4	36.6
5.0	103.5 ± 16.6	54.2 ± 1.3	47.6	50.2 ± 7.4	32.3 ± 4.5	35.7

**Table 4 ijerph-17-08230-t004:** Average fine PM removal efficiency of the ACF filter for different wind speeds at an ACF depth of 600 mm.

Wind Speed (m/s)	PM_10_			PM_2.5_		
	Before Filtration (μg/m^3^)	After Filtration (μg/m^3^)	Removal Efficiency (%)	Before Filtration (μg/m^3^)	After Filtration (μg/m^3^)	Removal Efficiency (%)
0.1	119.4 ± 9.9	85.7 ± 5.2	28.2	62.6 ± 0.8	48.2 ± 3.4	23.0
0.5	103.4 ± 5.8	71.5 ± 5.4	30.8	55.5 ± 3.8	42.0 ± 4.5	24.4
1.0	109.8 ± 12.4	66.1 ± 0.4	39.8	56.2 ± 2.1	38.1 ± 5.1	32.2
2.0	120.7 ± 7.3	57.9 ± 5.7	52.0	58.2 ± 3.7	32.6 ± 3.7	44.0
3.0	110.0 ± 10.6	55.9 ± 14.7	49.2	52.8 ± 6.0	30.6 ± 7.3	42.1
5.0	136.4 ± 5.2	46.5 ± 3.1	65.9	66.5 ± 4.7	32.0 ± 5.7	51.9

**Table 5 ijerph-17-08230-t005:** Differential pressure characteristics of the ACF for different wind speeds.

Wind Speed (m/s)	Pressure Loss (Pa)		
	200 mm	400 mm	60 mm
0.1	2.2	1.2	2.1
0.5	29.3	1.7	2.6
1.0	27.0	10.2	19.3
2.0	71.2	26.5	50.2
3.0	137.9	45.2	114.3
5.0	188.9	98.9	146.0

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
