# Peer review of "Determination of the Optimum Removal Efficiency of Fine Particulate Matter Using Activated Carbon Fiber (ACF)"

_ijerph, 2020, doi:10.3390/ijerph17218230_

Round 1
Reviewer 1 Report
Congratulations on the completion of your study described in the manuscript “Determination of the optimum removal efficiency of fine particulate matter using activated carbon fiber (ACF)”. I have carefully read your draft paper and concluded that your study is useful and interesting and may be acceptable for publication after some major revisions are successfully completed. I like and enjoyed reading your draft paper, but I have a few questions and concerns with your work as presented, which I invite the authors to address or explain, and which are detailed below.
Sincerely, Reviewer.
- The address of the 3 authors is the same. Therefore, it is not necessary to use the numbers "1", "2" and "3". It would only be necessary to write the number "1" with the address and then write the 3 emails.
- Try to write in an impersonal way, avoiding the first person when writing. For example, on line 84 "we used..."; line 210 “we examined…”; line 220 “We examined…”, etc. Please review the entire document taking into account this consideration.
- Lines 128-132 contain information that should be placed in the methodology section. The results section should not contain information about how the experiment was carried out.
- Divide Figure 3 into two parts: Fig 3. a. PM 10; b. PM 2.5. In addition, try to place the figure 3 on the same page.
- Be careful with the international system of units. For example, in figure 3 the word "sec" appears. If you refer to "seconds" you must write them with the letter "s". Please review the entire document taking into account this consideration.
- The images in figure 4 are of poor quality. Try replacing them with other images. In addition, the word "Simgokcheon" is underlined.
- The author contributions section contains "xxx" in it. I understand that the names of the authors of the article must appear.
- The conclusion section is very poor. Try to make the conclusions of this paper more concrete.
- It is necessary to review the format of the references. There are many errors. Please note that the reference of a paper must contain: Authors, title, year, issue, pages, doi. Some of this information is missing in references. In addition, different errors appear. For example, reference 18 contains: "P ark, J.". Please review the entire document taking into account these considerations.
Please note that the comments are intended merely to assist the authors in improving the paper and ensuring that published papers are of the highest quality. They are in NO WAY intended to discourage or demean the authors personally

Author Response
"Please see the attachment."

Reviewer 2 Report
After reading the manuscript, I am confused about one thing. The introduction gives the impression that the railway tunnel is the focus of the current study, except it is not. As then the manuscript discusses lab testing of ACF in removing PM, which is followed by an unclear description of roadside measurements (of wind speed and PM). Wind speed seems to be a key parameter in the removal efficiency of PM. However, there is no discussion on how wind speed is measured (height, sampling frequency, instrument accuracy, etc.). In meteorology, it is common to measure wind speed at 10 m height. If not measured at 10 m, approximate corrections are made to estimate wind speeds at any given height. It is not clear from the manuscript why and how the authors chose to run the experiments within 0.1 to 5 m/s. Are these realistic values? What is the level of turbulence as the train passes at high speed in tunnels that results in PM scattering? Are these results, which were obtained using experimental conditions that allowed wind speed to vary between 0.1 to 5 m/s adequately represent railway tunnel conditions? From one figure (Fig. 4), the sampling point where wind speed measurements were taken seems to be at low height. Can one measure a wind speed to 3 to 5 m/s at such low height?
For specific issues/comments, please see the attached file.

Author Response
"Please see the attachment."

Reviewer 3 Report
To Authors:
Overall, this is a very intesresting paper in the aspect of filter the pollutants before reach the subjects and cause morbidity and subsequent mortality.
I do have some quetions that I would like answered.
Major points:
While I understand that the “ACFs” are with a diameter of ≤ 10 μm it did not say the exact diameter.
Authors should provide how the above-mentioned ACFs can filter the PM2.5 which the focus of the study if the ACFs diaeter is around 10 µm?
Is ACF feasible for filtering the PM in heavy windy or no windy days. Have the authors evaluated on this aspect?
Is it economically sounds to install on the roadsides and railtracksides?
Authors should provide the usage of ACF during rainy or winter season in the respect of PM removal efficiency.
Minor point:
Author contributions are incomplete.
Author Response
"Please see the attachment."

Round 2
Reviewer 1 Report
Dear authors,
Thank you for considering my comments on this paper. The article has acquired a better appearance, in terms of format and content.
Author Response
Dear reviewer
Thank you for your comments on my thesis.
And thank you again for your good evaluation.
I will continue to conduct related research in the future.
Thank you in advance.
Sincerely
Reviewer 2 Report
The revisions are satisfactory. Some grammatical and consistent use of PM10 or PM2.5 needed as in many cases 10 and 2.5 are not subscripted. I suggest authors include a section that discusses the limitations of the study.
Author Response
Dear reviewer
Thank you for your comments on my thesis.
And thank you again for your good evaluation.
First, with respect to the comments you commented on,
For matters requiring individual presentation of PM10 or PM2.5, the entire thesis has been revised. In addition, PM10 and PM2.5 have been unified as PM.
Also, secondly, in relation to the comments you commented on,
Research limitations have been added to lines 263-266 in "4. Conclusion and discussion". The limitation of this study is that only one site where filters can be applied is reviewed, and in the future, we plan to review and compare and analyze underground spaces and roadsides in consideration of various environmental factors.
I will continue to conduct related research in the future.
Thank you in advance.
Sincerely